# Stress-Induced Depression and Alzheimer’s Disease: Focus on Astrocytes

**DOI:** 10.3390/ijms23094999

**Published:** 2022-04-30

**Authors:** Oleg V. Dolotov, Ludmila S. Inozemtseva, Nikolay F. Myasoedov, Igor A. Grivennikov

**Affiliations:** 1Institute of Molecular Genetics of National Research Centre “Kurchatov Institute”, 123182 Moscow, Russia; dolotov@img.ras.ru (O.V.D.); lsi@img.ras.ru (L.S.I.); nfm@img.ras.ru (N.F.M.); 2Faculty of Biology, Lomonosov Moscow State University, Leninskie Gory, 119234 Moscow, Russia

**Keywords:** stress, depression, neurodegeneration, Alzheimer’s disease, astrocytes, antidepressants, brain derived neurotrophic factor, TrkB receptor

## Abstract

Neurodegenerative diseases and depression are multifactorial disorders with a complex and poorly understood physiopathology. Astrocytes play a key role in the functioning of neurons in norm and pathology. Stress is an important factor for the development of brain disorders. Here, we review data on the effects of stress on astrocyte function and evidence of the involvement of astrocyte dysfunction in depression and Alzheimer’s disease (AD). Stressful life events are an important risk factor for depression; meanwhile, depression is an important risk factor for AD. Clinical data indicate atrophic changes in the same areas of the brain, the hippocampus and prefrontal cortex (PFC), in both pathologies. These brain regions play a key role in regulating the stress response and are most vulnerable to the action of glucocorticoids. PFC astrocytes are critically involved in the development of depression. Stress alters astrocyte function and can result in pyroptotic death of not only neurons, but also astrocytes. BDNF-TrkB system not only plays a key role in depression and in normalizing the stress response, but also appears to be an important factor in the functioning of astrocytes. Astrocytes, being a target for stress and glucocorticoids, are a promising target for the treatment of stress-dependent depression and AD.

## 1. Introduction

According to Hans Selye, “stress is the nonspecific response of the body to any demand” [1]. Stress is also defined as a state of threatened (or perceived as threatened) internal dynamic balance (“homeostasis”) caused by external or internal stimuli (“stressors”) [2]. To achieve homeostasis, the highly conservative regulatory neuroendocrine system, the “stress system”, is activated through synchronized interaction between the hypothalamic–pituitary–adrenal axis (HPAA) and the autonomic nervous system [2]. In principle, stress is necessary to adapt to changing environmental or internal conditions and increase the chance of survival. It is known that moderate stress is able to activate mental and behavioral processes to find solutions to the challenges facing an individual. However, excessive and/or prolonged stressors and the consequent chronic deregulation of the stress system can lead to a wide range of chronic pathological conditions, including pathologies of the cardiovascular, endocrine, immune and nervous systems. One of the stress-related pathologies of the nervous system is depression (major depressive disorder). Currently, depression is a most widespread mental disorder worldwide, which, according to World Health Organization, affects approximately 280 million people in the world, or about 4.0% of the population, including at least 5% of adults [3]. The incidence of depression increases with age, so it is about 27% in the age group of 75–80 years, 33% in the age group of 81–85 years and reaches 46% in the age group of 91 years and older [4]. Considering that the frequency of neurodegenerative diseases (in particular, dementia) also increases with age and that depression often manifests itself in dementia, it is possible that pathological changes in the course of dementia are associated with the development of depression.

Depression, as a mental disorder, is characterized by two core symptoms, depressed mood and loss of interest or pleasure in nearly all activities, and may be accompanied by other symptoms such as cognitive impairments, sleep disturbance, psychomotor retardation or agitation, feelings of worthlessness or excessive or inappropriate guilt [5]. Depression significantly worsens the quality of life. A large percentage of suicides, especially among young people, is associated with depression.

Figure 1 schematically shows the general sequence of events leading to the development of chronic stress, depression and, ultimately, to the degeneration of nerve cells.

Among the main factors leading to the development of chronic stress and depression, the following should be noted:Chronic pathologies of the nervous and cardiovascular systems, as well as oncological diseases.Social and psychological factors related to human living conditions and contacts with surrounding members of society and the external environment.Excessive and prolonged intake of various pharmaceutical preparations, as well as toxic compounds from the external environment, the number of which increases with the deterioration of the overall environmental situation in the world.

Alzheimer’s disease (AD) is the most common cause of dementia, which is estimated to account for 60% to 80% of cases [6]. AD is considered one of the main causes of morbidity and mortality among the elderly [7]. The prevalence of AD in Europe is estimated at 5.0%, which is 3.3% in men and 7.1% in women [8]. AD is a slowly progressive brain pathology that begins many years before the onset of symptoms. Clinical symptoms in early stages of AD include difficulty remembering recent conversations, names, or events, which are often accompanied by apathy and depression. In later stages of AD, symptoms include impaired communication, disorientation, behavioral changes, and ultimately difficulty speaking, swallowing, and walking [6]. AD is characterized by the accumulation of beta-amyloid peptide (Aß) (amyloid plaques) in brain tissues and a destabilization of the cytoskeleton of neurons caused by hyperphosphorylation of microtubule-associated Tau-protein. However, the poor correlation between cognitive decline and amyloid plaques raises the question of whether Aß accumulation actually causes neurodegeneration in AD. The formation of neurofibrillary tangles of Tau correlates better with neurodegeneration and clinical symptoms, and although Aß can initiate a cascade of events leading to neurodegeneration, Tau hyperphosphorylation is assumed to be key in neurodegeneration in AD [9]. The vast majority of cases of AD belong to a sporadic form (usually, late onset of symptoms). The sporadic form of AD is associated with the interaction of genetic and environmental factors, and aging is the main risk factor. Two main genetic risk factors for sporadic AD have been identified. Firstly, the presence of the *APOE4* allele encoding one of the three isoforms of Apolipoprotein E (apoE2, apoE3 and apoE4), the main transporter of cholesterol in the brain, which is synthesized and secreted by astrocytes [10], is the most significant risk factor for sporadic AD [11]. Other genetic risk factors for sporadic AD are polymorphisms and mutations in a number of genes expressed in microglia, in particular, polymorphism in the TREM2 gene encoding transmembrane glycoprotein, which acts as a receptor on the surface of microglia and perceives lipids that are exposed after cell damage [12]. The familial form of AD (early onset) accounts for about 5% of cases of AD and is associated with mutations in the genes encoding the precursor protein for Aß and presenilins 1 and 2, which leads to increased aggregation of Aß, but a small part of mutations in the gene encoding presenilin 1 is not familial and occurs de novo [13].

The development of depression is accompanied by changes in the metabolism of nerve and glial cells and an impairment of synaptic transmission between neurons. Prolonged action of harmful factors can eventually lead to the degeneration of nerve cells in the brain (Figure 1). Recent ideas about the functions of astrocytes assign them an extremely important role both in the normal functioning of the brain and in the development of brain pathologies [14,15]. In particular, astrocytes are a source of neurotrophic factors, regulate synaptic transmission and neurotransmitter levels in the synaptic cleft [15,16,17], and regulate neurogenesis in the adult hippocampus [17,18]. Thus, astrocytes are key actors in the processes, the deregulation of which is considered as an important component of the pathogenesis of depression. Depression is a frequent symptom of AD and may precede the manifestation of AD symptoms. Stress is an important risk factor for depression [19,20], while there are no direct data on whether stress is a risk factor for AD. The etiology and mechanisms of development of both pathologies are obviously complex and unclear, but the question arises whether there are common features in relation to astrocytes and stress response. In this review, we briefly consider the central functions of astrocytes, the involvement of stress in depression, and the relationship between depression and AD. We review the data on atrophic changes in the brain in these pathologies and whether they affect astrocytes. The review also examines the effects of stress on astrocyte function in animal and in vitro models and the involvement of astrocytes in the development of depression and AD.

## 2. The Role of Astrocytes in the Functioning of Neurons

For a long time, it was believed that the human brain contains about 100 billion neurons and about one trillion glial cells (a ratio of 1:10). However, recent studies using more advanced cell counting methods have shown that the number of glial cells in the human brain is approximately equal to the number of neurons and ranges from 40 to 130 billion [21]. A characteristic feature of glial cells, both in the brain and on the periphery, is the lack of the ability to generate and conduct nerve impulses [22]. There are three main types of glial cells: astrocytes, microglia and myelin-producing cells (oligodendrocytes in the central nervous system (CNS) and Schwann cells in the peripheral nervous system). Astrocytes and oligodendrocytes are the most common type of glial cells in the CNS. For quite a long period of time, the main function of astrocytes was considered to be the passive support of neurons that transmit nerve impulses, process and store information in the brain. Currently, such views have been revised, and astrocytes are assigned an important role both in the normal functioning of the CNS and in the development of various pathologies [23,24,25].

In the brain, astrocytes are a structural component of the so-called tripartite synapse, which includes, in addition to astrocytes, pre- and postsynaptic endings of neurons [26,27,28]. The morphological complexity of these cells should be particularly noted. Recent data show that one mature rodent astrocyte covers from 20,000 to 80,000 μ^3^ of domain space in the brain, and at the same time can interact with 300–600 neuronal dendrites [29]. Moreover, morphological studies found that mature astrocytes are able to interact with many thousands of synapses and at the same time are able to unite with other astrocytes, occupying unique spatial regions in the brain [30]. Figure 2 illustrates some functions of astrocytes in the tripartite synapse.

The glutaminergic presynaptic neuron is shown as an example. The presynaptic terminal and postsynaptic neuron with astrocyte processes surrounding the synapse are shown. Glutamate is released into the synaptic cleft and activates iGluR on the postsynaptic membrane, facilitating further transmission of the nerve impulse. In addition, glutamate can also exert its effect through metabotropic receptors (mGluR) localized on the presynaptic membrane. Astrocytes remove excess glutamate from the synaptic cleft using the EEAT membrane transporter. Astrocytes express numerous neurotrophic factors (NTFS) that act through the corresponding receptors.

Among the most important functions of astrocytes, in addition to the structural support of neurons, the following should be noted:The elimination of neurotransmitters such as glutamate, gamma-aminobutyric acid, dopamine, norepinephrine from the synaptic cleft during the transmission of a nerve impulse using specific transporters. For glutamate, EEAT serves as a transporter [31].The regulation of the concentration of potassium ions in the synaptic cleft using specific inward K+ channels Kir 4.1. (for review see: [32]).The expression and secretion of neurotrophic factors regulated the functioning and the viability of neurons, such as BDNF (brain-derived neurotrophic factor), FGF (fibroblast growth factor), NGF (nerve growth factor), GDNF (glial cell-derived neurotrophic factor), etc. (for review, see [33]).

In addition, astrocytes modulate the functioning of surrounding neurons by releasing gliotransmitters such as glutamate, ATP (adenosine triphosphate) and D-serine from cells. Calcium ions play an important role in this process [34,35]. It has now been established that astrocytes are also involved in the regulation of the neurogenesis in the adult hippocampus [17,18].

The mechanisms by which astrocytes modulate neural homeostasis, synaptic plasticity and memory are still poorly understood. It is known that astrocytes form intercellular networks by interaction of connexin 30 (Cx30) and connexin 43 (Cx43) proteins of gap junctions. In double-knockout mice with Cx30 and Cx43, sensorimotor disorders and a complete lack of spatial learning and memory were revealed, which shows that astrocytic connexins and an intact astroglial network in the brains of adult animals are important for maintaining neural homeostasis, plasticity and memory formation [36]. In general, astrocytes are key regulators of processes occurring in the nervous system, impairment of which can be considered as important components of various pathologies of the CNS, including AD and depression.

## 3. Stress and Depression

Depression is a complex neuropsychiatric disorder characterized by various neuropathological and physiological symptoms [19]. The pathogenesis of depression remains unclear, and several hypotheses have been proposed for the mechanisms of development of this mental disorder [37]. In particular, the development of depression is explained by a deficiency of serotonin or norepinephrine in the synaptic cleft (monoamine hypothesis), a deficiency of neurotrophic factors in certain parts of the brain (neurotrophic hypothesis), an impairment of neurogenesis in the adult hippocampus (in particular, associated with a deficiency of neurotrophic factors and neuroinflammation), inflammatory processes (inflammatory/cytokine hypothesis) and the deregulation of the HPAA, leading to a long-term abnormalities in the levels of stress hormones.

The hypothesis of the deregulation of the HPAA as a basis for the development of depression is supported by clinical data on stressful events as a risk factor for depression, on HPAA hyperactivation in patients with depression and normalization of the HPAA with antidepressant therapy [19,37,38,39]. It is assumed that prolonged hyperactivation of the HPAA due to chronic stress, inflammatory processes and/or genetic predisposition may lead to an impairment of self-regulation of HPAA activity and a chronic increase in peripheral and central levels of CRF (corticoliberin), ACTH (adenocorticotropic hormone) and glucocorticoids (Figure 3). In turn, long-term elevated central and circulating levels of stress hormones cause changes in the functioning of the brain, leading to the development of depression. In particular, glucocorticoids effectively cross the blood–brain barrier (BBB), having a significant impact on the functioning of the brain [40].

Severe depression is often associated with hypercortisolemia, and glucocorticoid therapy can cause symptoms of depression [41,42]. Patients with depression have elevated levels of CRF in blood plasma, and increased expression of CRF was observed in postmorten samples of patients with a long history of affective disorders [39]. Experimental data indicate that an increase in circulating glucocorticoids and in brain CRF causes depressive-like behavior in rodents [39,43]. CRF-producing neurons of the paraventricular nucleus of the hypothalamus have projections not only into the median eminence of the hypothalamus, from where CRF is released into the bloodstream, connecting the hypothalamus with the pituitary gland, but also into various extrahypothalamic regions of the brain. It is assumed that elevated levels of CRF, which plays the role of a neurotransmitter or neuromodulator, are responsible for the development of depressive states in chronic HPAA hyperactivation [42]. In addition, in stress, CRF levels increase in the amygdala, and it is assumed that this leads to an increase in CRF levels in the hippocampus; both the hippocampus and the amygdala also contain significant populations of neurons synthesizing CRF [44]. In stress, CRF-producing amygdala neurons activate noradrenergic neurons of the locus coeruleus (LC), which is the main source of noradrenaline in the brain. In turn, noradrenergic neurons of the LC activate CRF-producing neurons of the paraventricular nucleus of the hypothalamus, which leads to HPAA activation. Glucocorticoids inhibit the expression of tyrosine hydroxylase in the LC [45]. Thus, chronic stress can lead to dysfunction of LC neurons, resulting in persistent deregulation of the HPAA, changes in brain norepinephrine and glucocorticoid levels and the development of depression [46].

There are accumulated data that allow us to consider the neurotrophin and stress hypotheses of depression as two sides of the same coin. Excessive levels of glucocorticoids or stress inhibit BDNF-mediated neuroplasticity in areas of the brain that process emotional experiences, that is, in the hippocampus and prefrontal cortex (PFC), while BDNF-mediated neuroplasticity increases in the nucleus accumbens (Nac), amygdala and ventral tegmental area (VTA) involved in the management of reward pathways [47,48]. Stress reduces the expression of BDNF and/or its high affinity receptor TrkB in the hippocampus and PFC, while BDNF increases in the Nac, amygdala and VTA. A decrease in BDNF-TrkB signaling causes the suppression of the functioning of neural networks in the hippocampus and PFC, including loss of synapses, while an increase in BDNF-TrkB signaling increases the activity of networks in the Nac, amygdala and VTA. The hippocampus and PFC are key regulators of HPA activity, providing negative feedback by activating glucocorticoid receptors (GR) expressed in these regions in large amounts [49]. It is assumed that depression is associated, in particular, with the reduced expression or activity of GR due to chronic stress, which leads to the deregulation of HPAA activity [38,48]. GR activity is regulated by BDNF-TrkB signaling, which suggests the involvement of BDNF in the regulation of stress response [48]. Interestingly, astrocytic GR are more sensitive to stress than neuronal ones, and, moreover, the development of depressive-like behavior in mice in the model of chronic social defeat stress is associated with reduced expression of GR in PFC astrocytes. Moreover, the absence of GR in astrocytes causes depressive behavior, but restoring of GR expression prevents the depressive-like phenotype [50]. It was demonstrated that the loss of astrocytes (not affecting neurons) in PFC is sufficient for the development of depressive-like behavior in rats, which indicates their key role in the development of this pathology [51].

Along with the deregulation of the HPAA, the presence of chronic low-grade systemic inflammation is associated with depression. A significant proportion of patients suffering from depression have elevated circulating markers of inflammation, such as inflammatory cytokines and C-reactive protein [52,53]. It is assumed that the desensitization of GR is an important cause of both HPAA deregulation and activation of the immune system [37,41], although there is no correlation between inflammation and glucocorticoid resistance in patients [54]. It is well-known that inflammatory cytokines are HPAA activators at all levels of the axis, and chronic inflammation may be the cause of HPAA deregulation. In addition, inflammatory cytokines can cause desensitization of GR by stimulating the expression of its inactive form [55]. There is evidence that peripheral inflammatory cytokines can activate the HPAA at the level of the hypothalamus, acting through circumventricular organs or vagus nerve afferents, or passing through the BBB when its functions are impaired [53]. An important point is that systemic inflammation and circulating inflammatory cytokines can cause neuroinflammation and even neurodegeneration by activating microglia [56].

Elevated brain levels of inflammatory cytokines lead to the development of depression, causing changes in synaptic levels of neurotransmitters and suppressing hippocampal neurogenesis [57]. There is evidence that chronic stress can disturb the integrity of the BBB, increasing the intake of circulating inflammatory cytokines to the brain [58]. In addition, although glucocorticoids exhibit well-known potent anti-inflammatory effects, their pro-inflammatory activity is also demonstrated (in particular, stimulation of the production of pro-inflammatory cytokines) [59]. In depression, pro-inflammatory effects may prevail with a corresponding simultaneous increase in the levels of both glucocorticoids and inflammatory cytokines [53]. Inflammatory cytokines can be produced by neurons, astrocytes and microglia. Both acute and chronic stress can (depending on the type and intensity) cause an increase in the levels of proinflammatory cytokines in the brain, and the central role in this is assigned to microglia, activated in particular by norepinephrine and glucocorticoids [60].

The stress-related activation of microglia and the subsequent neuroinflammation in the hypothalamus is an important prerequisite for the development of glucocorticoid resistance, HPAA deregulation and depression [61]. However, in stress, astrocytes can also be a source of increased levels of proinflammatory cytokines in a number of brain regions, including in the hippocampus. It has been shown that acute cold stress [62], acute stress caused by complex stressors combining psychological and physical components (water immersion with immobilization or “footshock stress”), causes an increase in levels of the key proinflammatory cytokine IL-1b in astrocytes, but not in microglia and neurons [63].

Hence, PFC astrocytes are directly involved in mood regulation, and when their functioning is disrupted due to stress, depressive states may arise. In addition, stress can cause primary inflammatory response in astrocytes, which can also contribute to the development of brain pathologies, such as depression.

## 4. Depression and AD

Clinical manifestations of AD include neuropsychiatric symptoms, which is a serious problem with this disease. The most frequent (49%) neuropsychiatric symptom in AD is apathy (i.e., loss of emotional reactivity and decreased motivation) [64], which is similar to one of symptoms of depression. The symptoms of depression are diagnosed in approximately 40% of cases, along with other frequent symptoms such as aggression, anxiety and sleep disorders [64,65]. Although the possibility of the effects of anti-AD pharmaceuticals on the development of symptoms of depression should be taken into account [66], it is obvious that neurodegeneration in AD causes disruption of the functioning of brain systems involved in mood regulation, which can lead to the manifestation of symptoms of depression. However, an important question is whether the development of depression may be a risk factor or a prodromal AD condition. A recent meta-meta-analysis found a more than three-fold increase in the risk of AD in clinically significant depression [56]. Interestingly, it was found that the peak incidence of depressive disorders is observed in patients with AD several years before and after the age of onset of dementia, which probably indicates the common neurobiological basis for the development of these pathologies [67]. It is assumed that HPAA deregulation, neuroinflammation and BDNF deficit in a number of brain regions may be common for the development of depression and AD [68,69]. Inflammation plays an important role in AD and manifests itself both in the form of neuroinflammation and in elevated levels of inflammatory factors in the blood [70]. The deregulation profiles of the HPAA in depression and AD have significant differences, consisting of constant hypercortisolemia in depression and increased circadian peak of cortisol in mild–moderate stages of AD [71]. Hypercotisolemia is observed in severe depression [72], while the risk of dementia increases with severe depression [73]. It is not clear how differences in the profiles of cortisol elevation in these pathologies are consistent with an increased risk of AD in depression.

An intriguing question is whether antidepressants have an effect on the time of the onset or progression of AD. The treatment of patients with mild-to-moderate AD with selective serotonin reuptake inhibitors (SSRIs) for 9 months showed a delay in cognitive performance decline [74]. These results are in line with another study, which showed that the treatment of depressed patients with AD for 2 years with SSRIs delayed the decline in cognitive functions and gray matter atrophy [75]. The treatment of depressed AD patients with vortioxetine, an atypical antidepressant with multimodal activity, showed an improvement in cognitive functions after 12 months compared to the control (subjects treated with common antidepressants) [76]. The treatment of patients with depression and mild cognitive impairment (MCI) with SSRIs for more than 4 years delayed the progression of from MCI to AD by approximately 3 years compared to antidepressant-free patients or compared with MCI patients without a history of depression [77]. In a Danish nationwide study [78], it was found that, including all patients who received antidepressants, the incidence of dementia was increased compared to the frequency among antidepressant-free individuals. However, continued long-term treatment with antidepressants of all classes was associated with a decrease in the incidence of dementia and, in particular, AD. Thus, the long-term use of SSRIs can improve the cognitive performance of depressed patients with AD or slow cognitive decline and even delay the onset of AD. However, an important disadvantage of antidepressants is that they may cause serious side effects and be a risk factor for mortality in AD patients [79].

## 5. Astrocytes Atrophy in Depression and AD

Since astrocytes play a crucial role in the normal functioning of neurons, the question arises whether astrocytes undergo changes during depression and AD and the consequences of chronic stress. Astrocytes express receptors for all stress hormones and therefore are a target for them both in normal stress response and in conditions of impaired self-regulation of HPAA activity. Neuroimaging data show that in depression there are noticeable structural changes in a number of brain regions—first of all, a significant decrease in the volume of the closely interconnected medial PFC and hippocampus [14,16,80]. Apparently, such changes accompany a long-term and serious pathology, since a decrease in the volume of the hippocampus is observed only in cases of depression lasting longer than 2 years or in repeated episodes of the disease [81]. The interaction between the medial PFC and hippocampus integrates motivation, attention, memory and the results of past actions as the relevant circumstances change, which ensures adaptive behavior and mental health [59].

Atrophy of these brain regions is considered as a significant predictor of the development of clinical dementia [45,82,83]. Consequently, atrophy of the medial PFC and hippocampus may be associated with pre-existing depression and/or with early stages of dementia development. A decrease in the volume of these brain regions in patients with depression may be associated with both a decrease in astrocyte density [84,85], and neuronal atrophy [85,86] and a decrease in the number and functioning of synapses [80,87]. This brings depression closer to neurodegenerative diseases. Data on the presence of changes in the density of astrocytes and their sizes in various brain regions in depression are contradictory [88]. However, in particular, a decrease in astrocyte density in the hippocampus in patients with depression [84,89] or who received chronic treatment with glucocorticoids [89] has been shown. It should be noted that there is a significant problem in the identification of astrocytes, because astrocyte marker GFAP (glial fibrillary acidic protein) used in most studies can identify not all, and sometimes only a small part of astrocytes in tissues [90], which can lead to significant difficulties in assessing real differences in astrocyte density in brain tissues in various pathological conditions. In addition, a decrease in GFAP expression in astrocytes during depression can potentially lead to a decrease in the number of detected GFAP-positive cells [85,88].

A recent study shows that a decrease in hippocampal volume is associated with significant neurodegeneration in the CA1 region of the hippocampus at advanced stages of AD [91]. Additionally, there are indications of astrocyte atrophy in patients with advanced stages of AD [92]. Interestingly, astrocytes obtained from induced pluripotent stem cells (IPSC) of patients not only with familial, but also with a sporadic form of AD (the only patient studied), also showed features of atrophy in vitro in comparison with control astrocytes [93]. With regard to depression, a recent study using astrocytes differentiated from IPSC of healthy and depressed donors (treated with SSRIs) did not reveal morphological differences between cell lines, as well as significant differences in the expression of astrocytic markers and astrocytic glutamate transporter (EAAT2), and in the transcriptome profile [94]. There are still insufficient data on whether there is a correlation between the clinical manifestations of sporadic AD or depression and atrophy or other changes in astrocytes obtained from iPSC. However, it has already been shown that IPSC-derived human astrocytes carrying *APOE* ε4/ε4 genotype are less efficient than those with *APOE* ε3/ε3 in neuronal survival and synaptic integrity [95] and in the uptake and clearance of Aβ [96]. Research in this direction may shed light on how genetic risk factors for AD affect the functioning of astrocytes. Considering that the most significant genetic risk factor for sporadic AD is associated with the expression of isoforms of apolipoprotein E, the main source of which in the brain are astrocytes, this could help to identify the influence on the development of AD of the interaction of environmental factors, such as stress, with a genetic predisposition to AD.

## 6. Astrocytes Atrophy and Death in Experimental Animal Models of Chronic Stress

In experimental models, chronic stress causes numerous changes in the brain and, in particular, leads to pathological changes in neurons, such as dendrite spine and synaptic loss [97,98]. It was shown that chronic unpredictable stress (CUS), which is a widely used naturalistic model of depression, causes the hyperphosphorylation of Tau-protein in the hippocampus and PFC of rats, the phenomenon critically involved in neurodegeneration in AD [60]. Chronic restraint stress causes a selective decrease in the volume of the hippocampus in rats [99]. Interestingly, in rats, acute stress increases the number of apoptotic cells in the hippocampus, but CUS reduces this number. At the same time, both acute and chronic stress reversibly inhibit proliferation, but not the migration, survival and neuronal differentiation of new cells in the hippocampus [100]. Chronic pain–emotional stress caused signs of neuron atrophy and reduced their number in the hippocampus of rats via a non-apoptotic pathway [101]. This is consistent with the finding that chronic social defeat stress led to the pyroptosis of neurons in the hippocampus of stressed mice [102]. Pyroptosis is a pathway of programmed inflammatory cell death regulated by specific pro-inflammatory caspases and is characterized by the activation and release of potent pro-inflammatory cytokines IL-1ß and IL-18 through newly formed membrane pores into the extracellular space [103]. Pro-inflammatory caspases are activated from inactive pro-caspases in a multimeric specialized structure termed inflammasome, in particular, NLRP3 (nucleotide-binding, leucine-rich repeat, pyrin domain containing 3) inflammasome [104]. Acute stress rapidly increases extracellular glutamate levels and, as a consequence, the release of ATP from astrocytes (and, possibly, neurons) and an increase in the active form of NLRP3 inflammasome in the rat hippocampus [105]. Extracellular ATP activates the P2X7 receptor, which reversibly forms channels permeable to hydrophilic solutes with a molecular weight of up to 900 Da, and stimulates the formation of active NLRP3 inflammasomes and a subsequent release of pro-inflammatory cytokines (for review, see [106]). Blocking of the P2X7 receptor reversed the depressive-like behavior in the CUS model, which indicates its involvement in depression [105]. Microglial cells predominantly express P2X7 receptors, but astrocytes also possess P2X7 receptors and are involved in depressive-like behavior caused by high-intensity stressors [107]. Consequently, astrocytes, being a source of stress-induced extracellular ATP and carriers of the P2X7 receptor, are key participants in stress-induced ATP-mediated neuroinflammation and depression.

Interestingly, in chronic social defeat stress, depressive-like behavior is associated with low levels of ATP in the PFC, and conversely, blocking the release of ATP by astrocytes induces this behavior, while the administration of ATP or stimulating endogenous ATP release from astrocytes has an antidepressant-like effect [108]. This may indicate a decrease in the functioning of astrocytes under chronic stress. The exact role of ATP and P2X7 receptors in stress and depression is not clear. In particular, it was found that acute and chronic repeated immobilization stress reduces the expression of P2X7 receptors in the hippocampus [109], which suggests that ATP-mediated neuroinflammation is a fairly regulated process.

The experimental modeling of stress-induced depression also indicates the atrophy of not only neurons, but also astrocytes in the chronic hyperactivation of the HPAA. Chronic psychosocial stress leads to a decrease in the size of the hippocampus in tree shrews due to a decrease in the density and size of astrocytes, and the clinically used antidepressant fluoxetine prevents these changes [110]. In rats, chronic restrained stress induced astrocyte atrophy in the PFC without decreasing in their number [111]. This study also indicates that a decrease in the number of GFAP-positive cells may not reflect a decrease in the number of astrocytes, but is caused by a decrease in GFAP expression in cells [111]. In the CUS paradigm, a predominant NLRP3-mediated pyroptotic death of astrocytes was observed in the mouse hippocampus [112].

Therefore, chronic stress can cause the death of not only hippocampal neurons, but also astrocytes, and the path of cell death is associated with the induction of neuroinflammation due to the release of pro-inflammatory cytokines into the extracellular space.

## 7. Markers of Changes in Astrocytes Functions in Experimental Chronic Stress, Depression and AD

Along with a decrease in astrocytes density, possibly occurring only in the most severe cases of depression, there are data indicating changes in the functioning of astrocytes in patients with mild depression. In the cortex, LC and hippocampus, transcriptomic analysis revealed a depression-associated decrease in the expression of astrocyte-specific glutamine synthetase and glutamate transporters, which are involved in glutamate-glutamine cycle between neurons and astrocytes and play an important role in the functioning of synapses [113,114,115,116,117]. Depression is also associated with the suppressed expression of astrocytic S100B [114], aquaporin 4 [85], as well as connexins 30 and 43 and the transcription factor sox-9 regulating the expression of connexins [118,119,120].

As for the expression of glutamate synthetase in AD, a decrease in the expression of this enzyme in the temporal cortex was found compared with controls [121]. Interestingly, in contrast to depression, increased protein expression of aquaporin 4 in the frontal cortex was found in AD [122], and hippocampal mRNA and protein expression of Cx43 (also known as GJA1) was found up-regulated in AD [123,124,125]. It is possible that such opposite differences in the expression of these astrocytic proteins between depression and AD are associated with different levels of astrocyte atrophy and/or astrocyte activation, more pronounced in AD, or reflect a compensatory mechanism in response to neurodegeneration in AD.

The results obtained with patient tissues, indicating a change in the transcription of genes involved in the regulation of astrocyte functions, are of very high value for understanding the role of astrocytes in depression. However, it is obvious that such data have serious limitations due to the difficulty of establishing causal relationships between the detected changes and the disease and taking into account factors such as individual differences, age and concomitant pathologies of patients. In this regard, the use of experimental models plays an important role in the study of the mechanisms of astrocytes involvement in the development of depression and AD. 

In CUS, astrocyte dysfunctions were detected, such as the impairment of glutamate transport and metabolism in the rat cerebral cortex [126]. CUS results in a decrease in the expression of the glutamate transporter GLT-1 at mRNA and protein levels in the rat hippocampus [127]. However, chronic immobilization stress increases GLT-1 expression in the rat hippocampus [128,129], which may reflect the adaptation of animals to a repetitive (predictable for animals) stressor.

Using the CUS model, a decrease in the expression of Cx43 and an impairment of astrocytic intercellular contacts were detected in the rat PFC. The administration of selective serotonin reuptake inhibitors (i.e., antidepressants) or a GR antagonist (mifepristone) prevented or reversed these changes [130]. The prolonged administration of corticosterone to mice also caused an antidepressant-reversible decrease in Cx43 expression in the hippocampus [131]. It is important to note that the administration of glucocorticoids to experimental animals does not fully reproduce the effect of stress, in particular, on the functioning of synapses, and it is assumed that, under stress, there is a synergistic effect of glucocorticoids and other stress hormones, such as CRF and norepinephrine [14,132].

In pathological brain conditions, astrocytes undergo a number of functional and morphological changes, passing into a state called reactive astrocytes. Such astrocytes have an increased thickness of processes and begin to express nestin, which is a marker of nerve progenitor cells, and overexpress two other protein components of the astrocyte cytoskeleton, GFAP and vimentin [133]. In AD, the transition of astrocytes to a reactive state is initiated by the activation of microglia and the formation of amyloid plaques and tau-tangles. Reactive astrocytes contribute to the development of neuroinflammation by releasing inflammatory cytokines, nitric oxide, ATP and reactive oxygen species (for review, see [134]). Obviously, the reactive state of astrocytes develops during the progression of AD. Moreover, if in the serious stages of AD, there is an increase in GFAP expression in hippocampal dentate gyrus; then, in the early stages, there is a decrease in GFAP expression [135]. Remarkably, a similar situation occurs in depression and chronic stress when there is a decrease in GFAP expression in the hippocampus and cortical areas [85,136,137], suggesting a decrease in the functioning of astrocytes both in depression and in the early stages of AD.

The increased expression of GFAP in the AD brain is reflected in the elevated levels in the cerebrospinal fluid (CSF) [138]. In accordance with the involvement of reactive astrocytes in a wide spectrum of brain pathologies, increased GFAP CSF levels were detected in other types of dementia [138]. Interestingly, blood GFAP demonstrated a high performance in distinguishing individuals with AD and with other forms of dementia (for review, see [139]), and significantly outperformed CSF GFAP in detecting the activation of astrocytes and Aβ pathology in the early stages of AD [140].

Surprisingly, despite a decrease in GFAP expression in the brain in depression, it has recently been found that GFAP levels in both CSF [141] and blood [142] are also elevated in depressed patients. Moreover, blood GFAP levels increased with the severity of depression [142]. As mentioned above, chronic stress can cause not only the atrophy of hippocampal astrocytes, but also pyroptosis. In this regard, it is unclear whether the increase in the levels of the intracellular astrocytic protein GFAP in the blood and CSF is a consequence of the death of astrocytes possibly occurring in AD and depression. On the other hand, GFAP is present in the blood and CSF of healthy individuals [141,142]. This suggests that the release of GFAP into extracellular space, CSF, and, finally, into the blood may be a normal process that increases in depression and AD, and that GFAP may have extracellular functions. In this regard, it was found that another cytoskeletal protein, vimentin, has many non-mechanical intra- and extracellular functions and, in particular, promotes axonal growth and shows a neurorepairing effect (for review, see [143]).

In general, it seems that there is a significant similarity in the expression of the astrocytic marker GFAP in the brain and its levels in body fluids in depression and early stages of AD. It would be useful to know if there is the same similarity for other astrocytic biomarkers. In this regard, the data for the glycoprotein biomarker of neuroinflammation YKL-40, expressed in various cell types and involved in survival, proliferation and differentiation (see for references [144]), would be promising. In the human brain, YKL-40 immunoreactivity was found only in astrocytes, and the number of YKL-40-positive cells increases in several neurodegenerative diseases, including AD [145]. YKL-40 levels in CSF are also elevated in a number of neurodegenerative diseases, and in AD, YKL-40 levels are elevated already in the early preclinical stages (for review see [146]). It is not yet known whether YKL-40 increases in depression and whether antidepressant therapy affects its levels.

## 8. Effects of Stress Hormones on Astrocyte Function In Vitro

Despite the obvious limitations of in vitro studies, cell cultures are an extremely valuable tool for studying the role of astrocytes in the functioning of the brain, the development of its pathologies and the effects of different treatments. The acute effects of glucocorticoids on astrocytes function are currently the most well studied. Dexamethasone (a GR agonist devoid of mineralocorticoid activity) has been shown to increase the levels of the glutamate transporter GLT-1, but not GLAST (glutamate-aspartate transporter) in the cultures of primary astrocytes derived from the rat brain cortex [147]. The acute administration of corticosterone stimulated the expression of FGF-2 in rat cortical astrocytes [148,149,150]. At the same time, the expression of the neurotrophic factor NT-3 also increased, but the expression of BDNF and NGF decreased [148]. Dexamethasone administration reduced the expression of insulin-like growth factor-1 (IGF-1) [151] and NGF, as well as S100B, in astrocytes derived from the rat cerebral cortex, but stimulated the expression of FGF-2 in astrocytes from the hippocampus [152,153] and cortex [154]. Dexamethasone reduced the production of GDNF and increased the production of pro-inflammatory cytokine IL-2 in rat cortical astrocytes, while fluoxetine normalized the production of IL-2 but not GDNF [155]. In the culture of the rat astrocytoma C6, which shows significant similarity with primary rat astrocytes at late passages [156,157], dexamethasone reduced the expression of VEGF [158] and IGF-1 [159], and the production of GDNF [160]. Therefore, glucocorticoids in both C6 cells and rat primary astrocytes in vitro are suppressors of the expression of a number of key neurotrophic factors but stimulators of the inflammatory cytokines. Corticosterone reduced the biosynthesis of Cx43 in the astrocytes of the rat cortex and hippocampus and stimulated its biodegradation [161]. Prolonged exposure of dexamethasone in primary mixed astrocytes–oligodendrocytes cultures obtained from cerebral cortex of the rat also reduced Cx43 levels [162]. In general, this is consistent with the effects on astrocytes obtained in animal models of stress.

Interestingly, a transcriptomic analysis of the response of astrocytes obtained from IPSC from non-depressed donors to acute or chronic administration of cortisol in vitro revealed no changes in the expression of the above neurotrophic factors or their receptors, connexins and glutamate transporters [94]. Both the acute and chronic administration of cortisol reduced the expression of a number of proinflammatory cytokines in these astrocytes, in contrast to the effect of dexamethasone on primary cortical rat astrocytes [155]. This discrepancy may be due to both significant differences between human and rodent astrocytes [163,164], and insufficient maturity of astrocytes obtained from IPSC, since the expression of astrocyte markers in them is similar to embryonic astrocytes [94]. Perhaps this model most fully reflects the properties and susceptibility of astrocytes to various influences during the early development of the human brain [94]. An analysis of 334 differentially expressed genes under chronic cortisol action revealed the most pronounced changes in the expression of genes associated with regulation of cell adhesion, tyrosine kinase signaling, extracellular matrix organization, gliogenesis and positive regulation of cell death [94].

The effects of another stress hormone, CRF, on astrocyte functions have been less studied. Activation of the low-affinity CRF receptor (CCR2) stimulated NGF production in astrocytes derived from rat hippocampus [165]. CRF, acting via the high affinity receptor CRHR1, increased, unlike glucocorticoids, the production of Cx43 in cultured astrocytes and enhanced intercellular communication [166].

The effects of norepinephrine on astrocytes have also not been studied in detail yet. However, recent investigations showed that norepinephrine reduced the expression of the glutamate transporter GLT-1 in rat spinal cord astrocytes in vitro [167], but increased mRNA expression and production of BDNF, NGF, GDNF, FGF-2 and anti-inflammatory IL-6 in rat cortical astrocytes, acting via beta2-adrenergic receptors [168]. It was shown that the activation of beta2-adrenergic receptors in astrocytoma C6 cells in vitro stimulated the expression of NGF, and dexamethasone enhanced the agonist effect, showing a synergistic effect [169]. In this regard, it can be expected that stress hormones can modulate the influence of each other on the function of astrocytes. The combined effect of stress hormones on astrocytes is close to what happens during the body’s response to stress. However, these combined effects remain practically unexplored in cell culture models. It is obvious that the acute effects of stress hormones can be radically different from chronic effects. However, there is still insufficient information about the chronic effects of stress hormones on astrocytes. In general, the currently available data on the effects of stress hormones on astrocytes in vitro are in accordance with the data obtained in animal models and in the study of tissues of patients with depression and AD.

## 9. Effects of Antidepressants on Astrocytes In Vitro

Studies using cell cultures show that clinically applied antidepressants have significant effects on the functions of astrocytes. Selective serotonin reuptake inhibitors were found to stimulate glucose metabolism and expression of neurotrophic factors BDNF, VEGF and VGF in mouse cortical astrocytes, but not FGF-2, GDNF and IGF-1, while tricyclic antidepressants showed no effects [170]. However, the tricyclic antidepressant imipramine stimulated the expression of GDNF [171] and BDNF [172] in rat cortical astrocytes. Another tricyclic antidepressant, amitriptyline, also stimulated the expression of BDNF in rat cortical astrocytes [173]. Both selective serotonin reuptake inhibitors (fluoxetine and paroxetine) and tricyclic and tetracyclic antidepressants, but not haloperidol and diazepam, stimulated the expression and secretion of GDNF by rat astrocytoma cells C6 [174]. However, in another study, haloperidol, as well as atypical antipsychotics, also stimulated the secretion of GDNF by C6 cells [175].

The expression of a number of neurotrophic factors, including GDNF and BDNF, is under the control of the transcription factor CREB (cAMP responsive element binding protein). Antidepressants of various classes, but not haloperidol, increased GDNF secretion and activate CREB in C6 cells during acute and chronic (3 days) administration [176]. Amitriptyline, when administered acutely, increased the levels of CREB activation in C6 cells and astrocytes obtained from human embryos [177]. Interestingly, acute fluoxetine stimulated the release of ATP from rat and mouse hippocampal astrocytes in vitro and subsequently increased BDNF in astrocytes [178], while extracellular ATP transiently increased activation of CREB and expression of BDNF in cultured rat cortical astrocytes [179]. Thus, the administration of antidepressants stimulates the expression of key neurotrophic factors in cultured astrocytes and C6 cells.

Data on the effect of antidepressants on connexins expression and the effectiveness of intercellular contacts are quite contradictory. Fluoxetine increased Cx43 mRNA and protein levels in human astrocytoma cells [180]. Amitriptyline stimulated the expression of Cx43 and increased the efficiency of intercellular communication in cultured rat cortical astrocytes [181]. However, a study of the effects of antidepressants of various classes on mouse cortical astrocytes revealed the absence of a stimulating effect on the expression of Cx43 and a decrease in the effectiveness of intercellular contacts, or the absence of an effect for all antidepressants except paroxetine [182]. In this regard, the question of the effects of antidepressants on the functionality of intercellular contacts in cultured astrocytes requires further investigation.

Thus, cell culture models show that astrocytes are a target for antidepressants. Antidepressants regulate their functionality and, in particular, selectively stimulate the expression of neurotrophic factors. However, it is currently not clear what their effects are on astrocyte characteristics such as the functionality of neurotransmitter transporters and the expression of inflammatory mediators. It is also unknown how and in what ways antidepressants can modulate the effects of stress hormones on astrocytes.

## 10. Changes in the Levels of Neurotrophic Factors in the Brain in Depression and AD

A meta-analysis of studies of the levels of neurotrophic factors VEGF, GDNF, IGF-1 and IGF-2 in postmortem samples did not reveal their changes in AD [183]. Most of the analyzed studies demonstrated increased NGF levels and decreased BDNF levels in the hippocampus and neocortex (frontal cortex, temporal cortex and parietal cortex) [183]. Thus, in AD, there is a regulation of BDNF and NGF levels in the brain regions in which atrophy or death of neurons and astrocytes occur. In some of the studies, it was shown that there is an increase in the levels of immature NGF, but the pro-form of NGF that can cause apoptotic death of neurons through binding to p75 receptors [184]. It is unknown whether the mature or pro-form of BDNF decreases in AD, but it has been reported that both forms of BDNF decreased in the parietal cortex of AD patients [185]. The data suggest that AD is associated with a relatively selective alteration in the levels of brain neurotrophic factors, mainly expressed in a decrease in hippocampal and cortical BDNF levels.

Numerous experimental studies show that, in animal models, chronic stress results in a decrease in BDNF levels mainly in the hippocampus and PFC (see references in [186,187]). Despite the well-established role of BDNF in the development of depression and the effects of antidepressants [186,187], there are little data on changes in the levels of this neurotrophin in the brain of depressed patients. It was found that BDNF levels were reduced both in the PFC and in the hippocampus of suicide victims [188]. In another study, BDNF levels in PFC in elderly people with a history of depression did not significantly differ from BDNF levels in people without a history of depression, but it was found that BDNF was lower in the group of depressed elderly people with dementia than in patients with dementia and without depression [189]. Regarding the effect of antidepressants on BDNF in depression, elevated levels of BDNF in the hippocampus were found in patients with depression treated with antidepressants compared to non-treated patients [190]. This indicates that antidepressants increase BDNF levels in the hippocampus, but it is unknown how this compares with BDNF levels in healthy individuals. Another study found no significant differences in BDNF levels in the brain between healthy individuals and patients with untreated depression, but treatment with antidepressants significantly increased the level of BDNF in the parietal cortex [191]. Thus, there are indications that antidepressant treatment increases BDNF in the brain in patients with depression. However, there are still no data on the association of diagnosed depression with a decrease in the levels of brain BDNF, which may reflect a lesser extent of possible changes in the levels of BDNF in depression compared with AD. There are also no data on whether and to what extent brain BDNF levels change in severe and prolonged depression.

## 11. The Role of BDNF in the Morphogenesis and Functioning of Astrocytes

It is well-known that BDNF is the most important neurotrophic factor involved in the differentiation of CNS cells and their further functioning, including the morphological maturation of neurons and glia, and the formation of active synaptic contacts [192]. There are two types of receptors for BDNF that are expressed on the cells of the nervous system, including astrocytes p75 and TrkB. The p75 receptor participates in the regulation of glial cell proliferation and their response to various injuries [193]. Interestingly, astrocytes predominantly express a truncated form of the receptor (TrkB-T1), devoid of tyrosine kinase activity (Figure 4) [194,195].

It was shown that BDNF increased the viability of astrocytes by preventing their apoptosis caused by serum deprivation, and the anti-apoptotic effect of BDNF was prevented by a specific antagonist TrkB ANA-12 [195]. At the same time, BDNF induced activation of ERK, Akt and Src in astrocytes. Blocking of the ERK and Akt pathways canceled the protection of BDNF. In addition, BDNF protected astrocytes from death induced by of 3-nitropropionic acid (3-NP). This effect was also blocked by ANA-12, ERK and Src inhibitors. The conditioned medium of astrocytes treated with BDNF completely protected the neurons from 3-NP induced apoptosis [195]. This indicates that the neuroprotective effects of BDNF may involve indirect action via astrocytes.

Holt et al. [194] established the important role of the BDNF-TrkB-T1 signaling in the morphological maturation of astrocytes. The authors showed that mouse astrocytes in vitro express high levels of the BDNF receptor, TrkB, with almost exclusive expression of its truncated isoform, TrkB-T1 (90%). Moreover, the maximum level of expression was observed during the morphological maturation of astrocytes. It has been demonstrated that the morphological complexity of astrocytes increases in the presence of BDNF and depends on its signaling via TrkB-T1. The inactivation of TrkB-T1 in mice led to the appearance of morphologically immature astrocytes with significantly reduced cell volume, as well as to the impaired expression of a number of genes (Kir4.1, Aqp4, Glt1) characteristic of mature astrocytes and related to their functioning. Moreover, astrocytes with TrkB-T1 knocked out did not support normal synaptogenesis in neurons. These data indicate a significant role of the BDNF-TrkB-T1 system in the morphological maturation of astrocytes, a critical process for the development of CNS.

In a recently published paper [196], the authors demonstrated direct binding of typical antidepressants and ketamine (rapid antidepressant effect), with TRKB receptors, which led to their synaptic localization and effective activation. Mutations in the receptor site responsible for binding antidepressants eliminated both cellular and behavioral effects of antidepressants in vitro and in vivo. Thus, the authors proved that at least some effects of antidepressants can be carried out through direct regulation of activation of the BDNF-TRKB complex [196]. It is tempting to speculate that putative similar regulation of signaling through the truncated form of TrkB in astrocytes might be also involved in the clinical effects of antidepressants.

## 12. Possible Ways of Influencing Astrocytes Functions as Approaches to the Treatment of Depression and AD

As mentioned above, depression is one of the most common mental illnesses. By nature, depression is an extremely heterogeneous disease. Molecular and genetic factors, as well as environmental factors, contribute to its development. Unfortunately, there are no reliable biomarkers for diagnosing and predicting the course of various subtypes of depressive disorders [197]. This seems to be one of the reasons for the lack of positive effects from the use of approved therapeutic drugs in about 30% of patients with depression. Given the prevalence and severity of this disease, principally, new antidepressants are urgently needed. At the same time, the focus should be on identifying the triggers of this widespread and inherently heterogeneous disease. It seems impossible to find a single effective drug for the treatment of depression. However, considering that the deregulation of the stress system and reduced GR signaling are closely related to neuroinflammation, which is deeply involved in depression, much attention is currently being paid to the development of neuroimmune drugs for the treatment of mood disorders (for review, see [198]). It is noteworthy that one of the promising targets for the treatment of depression, the P2X7 receptor [198], is involved in astrocyte-mediated neuroinflammation, and is also considered a promising target for the treatment of AD [199]. Antidepressants increase the release of ATP suppressed in chronic stress by astrocytes and their production of neurotrophic factors, which supports the idea that the search for ways to normalize the functions of astrocytes may be promising for the treatment of depression and AD. Although the role of the BDNF-TrkB system in the pathophysiology of depression and AD is not completely clear, its involvement, at least in depression, is currently beyond doubt and is considered a promising target for the treatment of these pathologies [186,192,196,200,201]. Astrocytes seem to be not only a source of neurotrophic factors, but also a target for them, especially for BDNF, which could be used to normalize the functioning of astrocytes in depression and AD. However, the delivery of BDNF to the brain is associated with a complex of difficulties. Does the peripherally administered BDNF penetrate into the brain, and to what extent is this a debatable question [202,203]? In addition, if BDNF crosses the BBB by itself or with the help of carriers, an indiscriminate and uncontrolled increase in BDNF levels in the brain may initiate serious side effects, for example, epileptic activity [203,204]. One of the most promising alternative approaches may be the development of compounds capable of crossing the BBB and stimulating endogenous expression of BDNF and/or TrkB or enhancing of TrkB signaling in the brain, similar to what antidepressants and, possibly, short non-corticotropic fragments of the stress hormone ACTH and their analogues do [205,206,207,208]. Remarkably, antidepressants demonstrate a highly selective increase BDNF levels only in certain areas of the brain [209]. Another promising approach, in our opinion, is associated with the use of short peptide mimetics of BDNF, which are able to activate TrkB receptors demonstrating a pronounced antidepressant effects [210]. Taking into account the fact that there are direct interactions between the signaling pathways of glucocorticoids and BDNF [48], the normalization of BDNF-TrkB signaling in the hippocampus and PFC could lead to the normalization of HPAA activity, deregulated in depression and AD.

## 13. Conclusions

Clinical and experimental data suggest that stressful events are an important risk factor for depression. In turn, depression is not only a frequent symptom of AD, but can also be an important risk factor for AD. This suggests that stress response disorders may underlie both depression and AD. Clinical data indicate atrophic changes in the same areas of the brain, the hippocampus and PFC, in both pathologies. These same brain regions play a key role in regulating the stress response and are most vulnerable to the action of glucocorticoids due to the high expression of GR. The levels of GR in PFC astrocytes are critically important for the development of depression. There is clinical evidence that not only neurons, but also astrocytes undergo atrophy in depression and AD, although precise conclusions are difficult due to methodological problems in identifying astrocytes. Animal models demonstrate that chronic stress leads not only to atrophy, but also to the pyroptotic death of both neurons and astrocytes in the PFC. Clinical data indicate a decrease in the functioning of astrocytes in both in depression and AD. In animal and cell culture models, stress and glucocorticoids significantly alter the functions of astrocytes, including the expression of neurotrophic factors and glutamate transporters. Astrocytes are key players in stress-induced ATP-mediated neuroinflammation and depression. Clinical data show that AD is mainly associated with a decrease in BDNF levels in the hippocampus and a number of areas of the cerebral cortex. The BDNF-TrkB system not only plays a key role in depression and in normalizing the stress response, but also appears to be an important factor in the functioning of astrocytes. This suggests that compounds capable of stimulating the production of endogenous BDNF or the activation of TrkB-receptors in the PFC and hippocampus may be potentially useful for the treatment or prevention of depression and AD via protection of astrocytes. Conventional antidepressants and ketamine have this ability to stimulate the BDNF-TrkB system. It is not yet known whether ketamine can delay the development of AD. However, long-term treatment with SSRI antidepressants is associated with a reduced rate of AD and a delay in the onset of AD symptoms. This supports the idea that the overlapping neurobiological bases of depression and AD might provide common ways to treat or prevent these pathologies. Progress in understanding the development, progression and treatment of depression and AD, as well as resistance to stress factors, is possible with the identification and decoding of genetic and especially epigenetic changes in the human genome that contribute to their course [211,212].

## Figures and Tables

**Figure 1 ijms-23-04999-f001:**
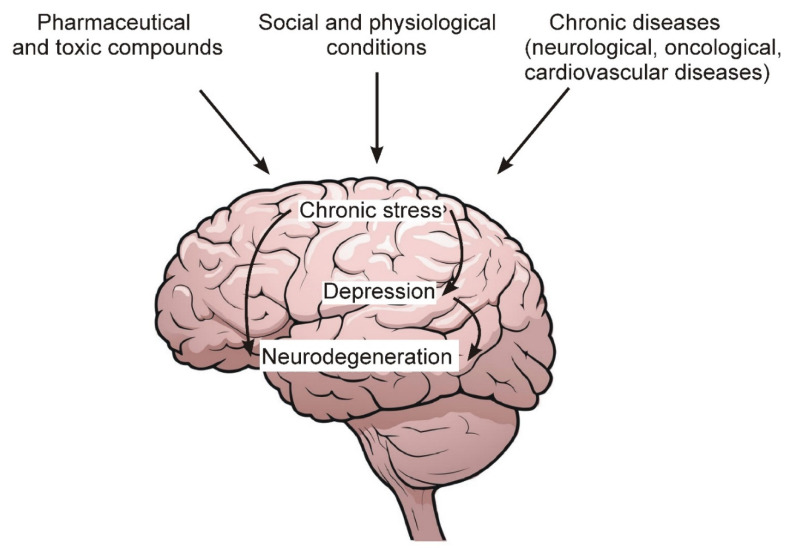
Some environmental factors, as well as chronic diseases that lead to the development of chronic stress and depression.

**Figure 2 ijms-23-04999-f002:**
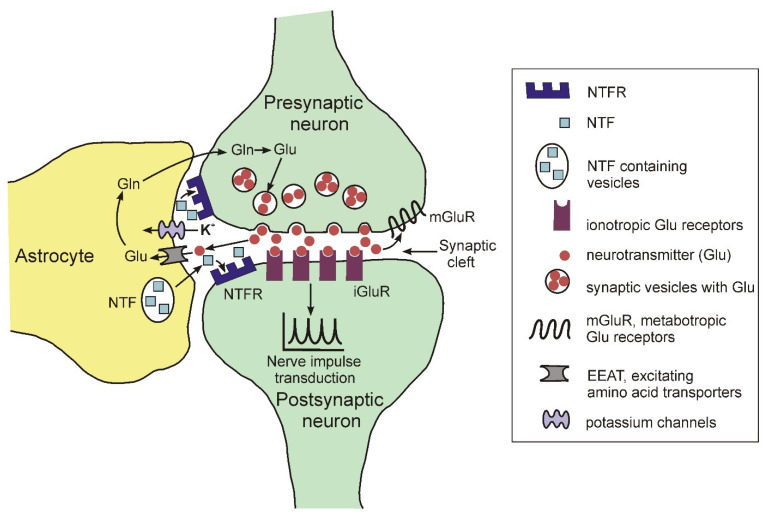
Schematic representation of the structure of the tripartite synapse.

**Figure 3 ijms-23-04999-f003:**
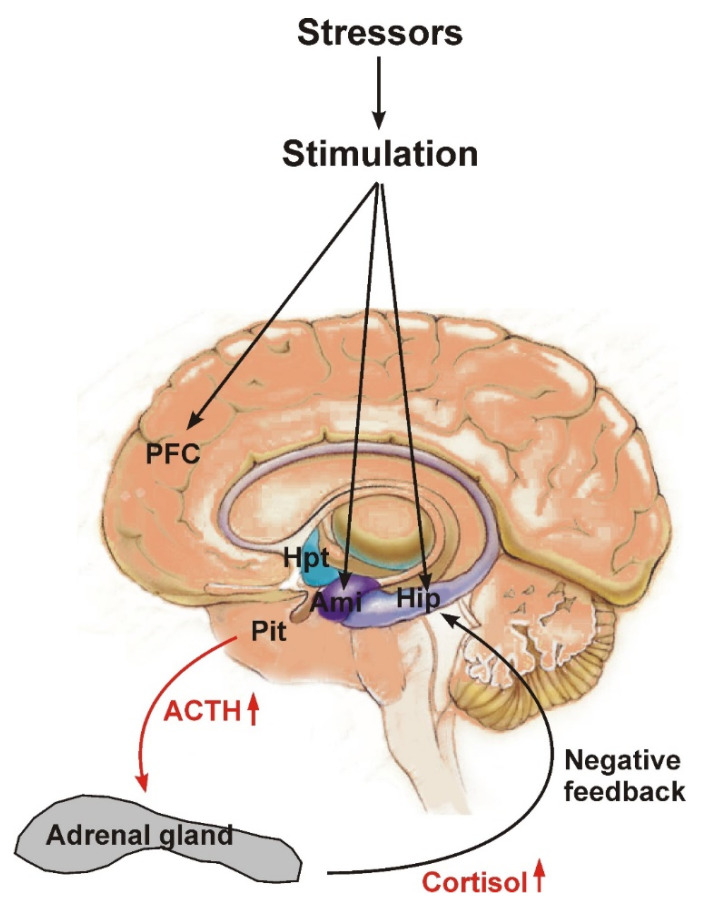
The hypothalamic–pituitary–adrenal axis (HPAA). The hypothalamus and higher brain centers such as the PFC, hippocampus and amygdala control HPAA activity. The release of CRF (corticotrophin releasing factor, corticoliberin) from the paraventricular hypothalamic area stimulates the release of adrenocorticotrophin hormone (ACTH) from the anterior pituitary, which, in turn, stimulates the release of glucocorticoids (cortisol in humans, corticosterone in rodents) from the adrenal cortex. The glucocorticoids secreted into the bloodstream, via glucocorticoid receptors (GR) expressed in the brain and pituitary, suppress HPAA increased activity through a negative feedback loop. The PFC and hippocampus play a key role in suppressing HPAA activity and express large amounts of GR, which makes these brain regions particularly sensitive to the damaging effects of glucocorticoids. Abbreviations: PFC = prefrontal cortex; Hip = hippocampus; Amy = Amygdala; Pit = Pituitary; Hpt = hypothalamus.

**Figure 4 ijms-23-04999-f004:**
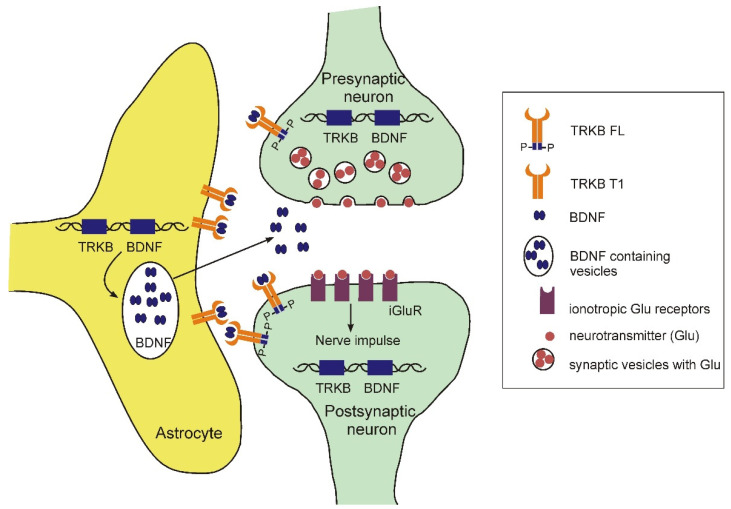
Schematic representation of the BDNF-TrkB system in the tripartite synapse. Some parts from Figure 2 are used in this scheme. On neurons, the main receptor for BDNF is the full-length form of TrkB (TrkB-FL) with tyrosine kinase activity, whereas on glial cells, there is mainly a truncated form of this receptor (TrkB-T1) that does not possess tyrosine kinase activity. Synthesis and subsequent release of BDNF from astrocytes leads to its binding to receptors on both neurons and astrocytes. Binding BDNF to TrkB-T1 on astrocytes leads to activation of certain signaling pathways involved in maintaining the viability and functioning of these cells (described in main text).

## Data Availability

Data are contained within the article.

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
