# Peer review of "Stress-Induced Depression and Alzheimer’s Disease: Focus on Astrocytes"

_ijms, 2022, doi:10.3390/ijms23094999_

Round 1
Reviewer 1 Report
In the article by Dolotov et al., titled “Stress-induced depression and Alzheimer’s disease: focus on astrocytes”, the authors discuss how stress can influence astrocytes during depression, how depression may precede Alzheimer’s disease pathogenesis in the elderly. The review is divided into 11 chapters focusing on the role of stress and growth factors, their role in disease pathogenesis in AD, and recent advances in the field. The review is timely, owing to global crisis and rise in population with depression. The review is lengthy, hard to concentrate on, and occasionally difficult to understand. Nonetheless, the following remarks have been made by this reviewer:
Line 43: Incorrect citation, moreover, WHO estimate is 280 million not 300.
Line 122: doi: 10.1007/978-981-13-9913-8_3 by Verkhratsky and Parpura should be cited.
Line 170: please change „ neuropsychic disease“ to neuropsychiatric disorder
Please re-check all citations (e.g Line 257, 263, 267: incorrect citations)
Lines 35-36: Large number must be changed.
Line 52: Symptoms
Lines 73-74: AD has two classifications Familial (early) and Sporadic (late), kindly rephrase the sentence accordingly.
Lines 78-79: Genetics is family history, the authors should mention genetic predisposition like ApoE, Trem.. as also contributing factors.
Lines 141-142 is repeated 148-149, and must be removed.
Lines 291-293: missing reference/s.
Line 322: remove the abbreviation for medial prefrontal cortex (mPFC) since it’s used only once in the text or use the mPFC where appropriate (326, 331..)
Lines 347-348: “To date, it is not clear what causes the decrease in the volume of the hippocampus in AD”, the sentence should be removed, and subsequent sentence rephrased. It is known that neuronal loss, in AD, is the reason for decrease in hippocampal volume.
Line 355: The authors mention IPSCs and cite Verkhratsky et al.2019, which is a review article and doesn’t deal with IPSCs and must be removed.
Lines 355-358: “Since obtaining IPSC… extremely valuable model is still unknown” is not necessary and can be removed.
Lines 373-375: “Chronic pain-emotional stress caused signs of neuron atrophy and reduced their number in the hippocampus of rats, but did not lead to an increase in the number of apoptotic cells” and cite Tishkina et al., 2009.The interpretation of the manuscript by the authors is wrong. The article report non-apoptotic loss of neurons and apoptosis is not the only mechanism for neuron loss.
Lines 375-376 and 377-378 are repeats.
Lines 403-408: In contrary, the increase may also represent compensatory mechanism/s against ongoing neurodegeneration.
Lines 406-408: oddly italic..
Lines 420-422: However “a study” .. and cite 2 references.
Line 451: “glucocorticoids in both C6 cells in vitro and rat primary astrocytes” should read “glucocorticoids in both C6 cells and rat primary astrocytes in vitro”.
Line 543: Numerous experimental studies..and cite only 2 articles. The sentence must be rephrased.
Lines 532-534: “Most of the analyzed studies demonstrated s and neocortex (frontal cortex, temporal cortex and parietal cortex)” missing citations.
Lines 552-554: This perhaps is not appropriate statement, since there are no concrete studies in this regard.
Lines 316-623: This statement is quite contradictory since BDNF has been demonstrated to cross BBB, see ref: Pan et al., 1998; doi: 10.1016/s0028-3908(98)00141-5. Also BDNF is strongly expressed in the brain, and its leak can be used to measure BBB damage (Lesniak et al., 2021).
Author Response
We are grateful to the reviewer for careful reading and very valuable comments. We have been able to incorporate changes to reflect the comments provided by the reviewer.
The manuscript has been corrected in accordance with all reviewer's comments. In particular:
Lines 73-74: AD has two classifications Familial (early) and Sporadic (late), kindly rephrase the sentence accordingly.
AA: We have rephrased the sentence (page 2 of the manuscript: “Clinical symptoms in early stages of AD…”).
Lines 78-79: Genetics is family history, the authors should mention genetic predisposition like ApoE, Trem.. as also contributing factors.
AA: We have included the relevant text (page 3: “The vast majority of cases of AD…”).
Lines 347-348: “To date, it is not clear what causes the decrease in the volume of the hippocampus in AD”, the sentence should be removed, and subsequent sentence rephrased. It is known that neuronal loss, in AD, is the reason for decrease in hippocampal volume.
AA: We have corrected this paragraph (page 9).
Lines 373-375: “Chronic pain-emotional stress caused signs of neuron atrophy and reduced their number in the hippocampus of rats, but did not lead to an increase in the number of apoptotic cells” and cite Tishkina et al., 2009.The interpretation of the manuscript by the authors is wrong. The article report non-apoptotic loss of neurons and apoptosis is not the only mechanism for neuron loss.
AA: We have corrected this sentence (page 10).
Lines 403-408: In contrary, the increase may also represent compensatory mechanism/s against ongoing neurodegeneration.
AA: We have added to the text this possibility (page 11).
Lines 552-554: This perhaps is not appropriate statement, since there are no concrete studies in this regard.
AA: We have corrected this part of the text (page 15: “However, there is still no data on the association of diagnosed…”).
Lines 316-623: This statement is quite contradictory since BDNF has been demonstrated to cross BBB, see ref: Pan et al., 1998; doi: 10.1016/s0028-3908(98)00141-5. Also BDNF is strongly expressed in the brain, and its leak can be used to measure BBB damage (Lesniak et al., 2021).
AA: We have corrected this statement (page 17: “However, the delivery of BDNF to the brain is…”).
We have corrected all other noted typos and errors.
Reviewer 2 Report
The manuscript entitled “Stress-induced depression and Alzheimer’s disease: focus on astrocytes” aims to review the central functions of astrocytes, the involvement of stress in depression, the relationship between depression and AD, and the data on atrophic changes in the brain in these pathologies and whether they affect astrocytes. The review also examines the effects of stress on astrocyte function in animal and in vitro models and the involvement of astrocytes in the development of depression and AD. The manuscript is well prepared, and this topic is of high importance since we witness the epidemic of AD and depression nowadays. The discussion on the mentioned hypothesis is done carefully and scientifically sound. Nevertheless, I have some small concerns and suggestions for improvement. Although the primary target for this manuscript is a pathophysiological connection between molecular targets of both diseases, the authors do not consider the role of depression therapy as a factor in AD development. Moreover, they mention that AD therapy may cause depression symptoms development. So, why not do it both ways? The emphasis on this important question would improve this manuscript significantly.
Author Response
Comment: Although the primary target for this manuscript is a pathophysiological connection between molecular targets of both diseases, the authors do not consider the role of depression therapy as a factor in AD development.
We are grateful to the reviewer for very valuable comment and advice. We have included the discussion of this important issue into the section “Depression and AD” (page 8: “An intriguing question is whether antidepressants have…”).
Reviewer 3 Report
Stress response disorders may underlie both depression and AD. Clinical data indicate atrophic changes in the same areas of the brain, the hippocampus and PFC, in both pathologies. These brain regions are most vulnerable to the action of glucocorticoids considered madiators of inflammation.
Animal models demonstrate that chronic stress leads not only to atrophy, but also to pyroptotic death of both neurons and astrocytes in the PFC.
BDNF – TrkB system not only plays a key role in depression and in normalizing the stress response, but also appears to be an important factor in the functioning of astrocytes.
Conventional antidepressants and ketamine have this ability to stimulate the BDNF – TrkB system.
BDNF does not cross the BBB, one of the promising approaches may be the development of compounds capable of crossing the BBB and stimulating endogenous expression of BDNF and/or TrkB or enhancing of TrkB signaling in the brain, similar to what antidepressants and, possibly, short fragments of the stress hormone ACTH and their analogues do.
Some suggestions to improve the revision
- Depression has not specific biomarkers, anyway precise biological treatments (aginst neuroinflammation but also regarding astrocytes) are under investigation in several trials worldwide. I suggets to improve the revision with some comments about this topic. Please cite and report the following review : « Drevets WC. Immune targets for therapeutic development in depression: towards precision medicine Nature reviews drug discovery 2022 ».
- Please report a paragraph (and maybe a table) with the main studies on GFAP in AD including the excellent review of « Abdelhak A et al. Blood GFAP as an emerging biomarker in brain and spinal cord disorders. Nature Rev Neurology 2022».
- Similarly, describe the relevance of YKL-40 (a further biomarkers of astroglia) in AD. In this regards the authors should refer and cite the following revisions : « Baldacci F. The neuroinflammatory biomarker YKL-40 for neurodegenerative diseases: Advances in development. Expert Rev. Proteom. 2019”; “Baldacci F. Diagnostic function of the neuroinflammatory biomarker YKL-40 in Alzheimer’s disease and other neurodegenerative diseases. Expert Rev. Proteom. 2017”.
-Sometimes authors write TkRB sometimes TrKB please correct accordingly.
Author Response
We are grateful to the reviewer for these valuable comments and advices. We have been able to incorporate changes to reflect the suggestions provided by the reviewer.
Comment:
Please report a paragraph (and maybe a table) with the main studies on GFAP in AD including the excellent review of « Abdelhak A et al. Blood GFAP as an emerging biomarker in brain and spinal cord disorders. Nature Rev Neurology 2022».
Similarly, describe the relevance of YKL-40 (a further biomarkers of astroglia) in AD. In this regards the authors should refer and cite the following revisions
Answer: We have included the relevant paragraphs into the section “Markers of changes in astrocytes functions in experimental chronic stress, depression and AD” (page 12).
Comment: Depression has not specific biomarkers, anyway precise biological treatments (aginst neuroinflammation but also regarding astrocytes) are under investigation in several trials worldwide. I suggets to improve the revision with some comments about this topic.
Answer: We have included the relevant comments into the sections “Possible ways of influencing astrocytes functions as approaches to the treatment of depression and AD” and “Astrocytes atrophy and death in experimental animal models of chronic stress” (page 17: “However, considering that the deregulation…” and page 10: “Extracellular ATP activates the P2X7 receptor, which reversibly forms channels permeable…”)